# Machine learning and natural language processing methods to identify ischemic stroke, acuity and location from radiology reports

**Charlene Jennifer Ong**[1,2,3,4]\*, **Agni Orfanoudaki**[4], **Rebecca Zhang**[4], **Francois Pierre M. Caprasse**[4], **Meghan Hutch**[1,2], **Liang Ma**[1], **Darian Fard**[1], **Oluwafemi Balogun**[1,2], **Matthew I. Miller**[1], **Margaret Minnig**[1], **Hanife Saglam**[3], **Brenton Prescott**[2], **David M. Greer**[1,2], **Stelios Smirnakis**[3], **Dimitris Bertsimas**[4,5]

1 Boston University School of Medicine, Boston, Massachusetts, United States of America, 2 Boston Medical Center, Boston, Massachusetts, United States of America, 3 Harvard Medical School, Boston, Massachusetts, United States of America, 4 Operations Research Center, Massachusetts Institute of Technology, Cambridge, Massachusetts, United States of America, 5 Sloan School of Management, Massachusetts Institute of Technology, Cambridge, Massachusetts, United States of America

\* cjong@bu.edu

**Data Availability Statement:** Radiology report data cannot be shared publicly because it consists of personal information from which it is difficult to

## Abstract

Accurate, automated extraction of clinical stroke information from unstructured text has several important applications. ICD-9/10 codes can misclassify ischemic stroke events and do not distinguish acuity or location. Expeditious, accurate data extraction could provide considerable improvement in identifying stroke in large datasets, triaging critical clinical reports, and quality improvement efforts. In this study, we developed and report a comprehensive framework studying the performance of simple and complex stroke-specific Natural Language Processing (NLP) and Machine Learning (ML) methods to determine presence, location, and acuity of ischemic stroke from radiographic text. We collected 60,564 Computed Tomography and Magnetic Resonance Imaging Radiology reports from 17,864 patients from two large academic medical centers. We used standard techniques to featurize unstructured text and developed neurovascular specific word GloVe embeddings. We trained various binary classification algorithms to identify stroke presence, location, and acuity using 75% of 1,359 expert-labeled reports. We validated our methods internally on the remaining 25% of reports and externally on 500 radiology reports from an entirely separate academic institution. In our internal population, GloVe word embeddings paired with deep learning (Recurrent Neural Networks) had the best discrimination of all methods for our three tasks (AUCs of 0.96, 0.98, 0.93 respectively). Simpler NLP approaches (Bag of Words) performed best with interpretable algorithms (Logistic Regression) for identifying ischemic stroke (AUC of 0.95), MCA location (AUC 0.96), and acuity (AUC of 0.90). Similarly, GloVe and Recurrent Neural Networks (AUC 0.92, 0.89, 0.93) generalized better in our external test set than BOW and Logistic Regression for stroke presence, location and acuity, respectively (AUC 0.89, 0.86, 0.80). Our study demonstrates a comprehensive assessment of NLP techniques for unstructured radiographic text. Our findings are

guarantee extraction. As a result, there is a possibility of deductive disclosure of participants and therefore full data access through a public repository is not permitted by the institutions that provided us the data. To facilitate data availability while maintaining confidentiality, we will make the data and associated documentation available under a data sharing agreement which includes: 1) commitment to using the data only for research purposes and not to identify any individual participant; 2) a commitment to securing the data using appropriate measures, and 3) a commitment to destroy or return the data after analyses are complete. Requests can be made to Megan C. Morash (mmorash@partners.org). Furthermore, we have provided examples of our radiographic text, methods, and code so that research can be done to reproduce our findings or use on external datasets. In particular, we have included an example of a radiology report and a link (http://www.mit.edu/~agniorf/files/Glove_Neurology_Embeddings.csv) to the neurology specific GloVe embeddings as a supporting information file. We also provide access to the full codebase of our analysis by sharing our GitHub repository (https://github.com/agniorf/StrokeDetectNLP).

**Funding:** The author(s) received no specific funding for this work.

**Competing interests:** The authors have declared that no competing interests exist.

suggestive that NLP/ML methods can be used to discriminate stroke features from large data cohorts for both clinical and research-related investigations.

## 1. Introduction

Radiographic findings on head computed tomography (CT) or magnetic resonance imaging (MRI) are frequently used to support or confirm the diagnosis of ischemic stroke in clinical practice. Radiologists interpret images in narrative reports that detail stroke occurrence and other pertinent information including acuity, location, size and other incidental findings. Because of their unstructured nature, radiology reports do not make it easy to employ these information-rich data sources for either large-scale, retrospective review, or for real-time identification of stroke in the clinical workflow. The ability to automate the extraction of meaningful data from radiology reports would enable quick and accurate identification of strokes and relevant features such as location and acuity. Such a system could help clinicians triage critical reports, target patients eligible for time-sensitive interventions or outpatient follow up, and identify populations of interest for research [1].

Natural language processing (NLP) is a field that spans multiple scientific disciplines including linguistics, computer science, and artificial intelligence. The main objective of NLP is to develop and apply algorithms that can process and analyze unstructured language. A distinctive subfield of NLP focuses on the extraction of meaningful data from narrative text using Machine Learning (ML) methods [2]. ML-based NLP involves two steps: text featurization and classification. Text featurization converts narrative text into structured data. Examples of text featurization methods include Bag of Words (BOW), Term Frequency-Inverse Document Frequency (TF-IDF) and word embeddings [2, 3]. Word embedding methods, including Word2-Vec and Global Vectors for Word Representation (GloVe) [3–5], learn a distributed representation for words. The result of these methods is a numerical representation of text that can be subsequently used for myriad applications. One particular medical application of these methods is the classification of salient findings from unstructured radiographic reports. After converting language into relevant binary or continuous features through text featurization, supervised classification models can separate reports into desired categories (i.e. presence or absence of acute middle cerebral artery stroke). These models are trained on a portion of the cohort, and then tested on unseen data to determine how accurately they classify observations. Previous efforts to automate diagnoses from radiologic text have resulted in algorithms that can identify pneumonia, breast cancer, and critical head CT findings [1, 3]. Specifically, Zech and colleagues found that simpler featurization and classification techniques perform comparably to more sophisticated deep learning approaches in identifying binary critical head CT classifiers (i.e. critical v. non critical; ischemia v. no ischemia) [1]. However, clinicians and radiologists use diverse language patterns to characterize stroke features. For instance, "sub-acute" is a relative term and can describe strokes that occurred anywhere from hours to months prior to the diagnostic study. Specific descriptions of ischemia on head CTs (i.e. hypo-densities or sulcal effacement) or MRIs (decreased Apparent Diffusion Coefficient (ADC)) provide clinicians with more context that allows them to infer timing, severity and likely diagnosis. We hypothesized that simpler NLP featurization approaches that rely on counting how many times a relevant word occurred in text, like BOW or tf-idf, may not sufficiently capture the language describing stroke features. Word-embedding approaches that account for word relationships might better identify characteristics of interest.

In this study, we aimed to: 1) expand the application of NLP to identify both the presence of ischemia and relevant characteristics including location subtype, and acuity; and 2) compare whether a neurovascular-tailored NLP featurization algorithm (GloVe) outperforms simpler methods (BOW, tf-idf) in identifying key qualifying characteristics.

## 2. Methods

### Study population

We collected 60,564 radiology reports consisting of head Computed Tomography (CT), or CT Angiography (CTA) studies, brain Magnetic Resonance Imaging (MRI), or MR Angiography (MRA) studies from a cohort of 17,864 patients over 18 with ICD-9 diagnosis codes of ischemic stroke (433.01, 433.11, 433.21, 433.31, 433.81, 433.91, 434.01, 434.11, 434.91 and 436) from 2003–2018 from the Research Patient Data Registry (RPDR), at Massachusetts General and Brigham and Women's Hospitals (Fig 1, S1 Table in S1 File) [6]. We chose these four imaging modalities because a generalizable algorithm that identifies stroke characteristics from multiple imaging report subtypes would have greater practical application. We externally validated our best performing classification methods on 500 radiographic reports from 424 patients who were admitted to Boston Medical Center between 2016–2018. Boston Medical Center is the largest safety-net hospital in New England, and thus has a markedly different racial-ethnic and socioeconomic population than our training cohort. The Partners Human Research Committee and Boston Medical Center local IRBs approved this study.

### Manual radiographic report labeling

1,359 original radiology reports from 297 patients (883 Head CTs or CTAs, 476 MRIs or MRAs) were hand-labeled by study team members trained by attending physicians and/or senior staff members (Fig 1). Each report included the text, type of scan (CT, MRI, CTA, or MRA), date, and report time (S1 Appendix in S1 File). Reports were distributed randomly among the labelers. Each reporter independently labeled 1) the presence or absence of ischemic stroke, 2) middle cerebral artery (MCA) territory involvement, and 3) stroke acuity. Stroke occurrence, acuity, and MCA location were classified as either present or absent. Labelers identified "stroke" if the report definitively reported a diagnosis of ischemic stroke or if ischemic stroke was determined as probable by the labeler based on the radiology report text. A stroke was labeled as acute if: the reporting radiologist reported it as acute in their report, diffusion restriction or apparent diffusion coefficient hypointensity without T2 prolongation was mentioned on MRI report, or it was interpreted as having occurred within the last 7–10 days. MCA stroke location was defined as a reported MCA territory or thrombus in MCA with corresponding symptoms in the history section of report. We focused on the identification of MCA stroke as this stroke subtype is particularly clinically actionable via thrombectomy and at high risk for stroke sequelae including edema and hemorrhagic transformation. Study data were collected and managed using a Research Electronic Data Capture (REDCap) electronic database [7]. Each report was separately labeled twice. Any discrepancies between the two labels were reviewed by attending neurologists CJO or SS. If labelers felt that identification of stroke occurrence or characteristics were indeterminate, they were labeled as absent. A board-eligible Attending Neurocritical Care physician (CJO) conducted a blinded analysis, and then adjudicated 300 radiology reports by review of images. In an assessment of 10% of the final reports labels from both the derivation and external cohorts by a trained physician and labeler (HS), percent agreement for stroke presence, MCA location, and acuity were 91%, 87%, and 93%, respectively, suggesting good to excellent inter-rater reliability. Additionally, a board-certified Neurologist and Neurointensivist (CJO) assessed the percent agreement of 300 reports

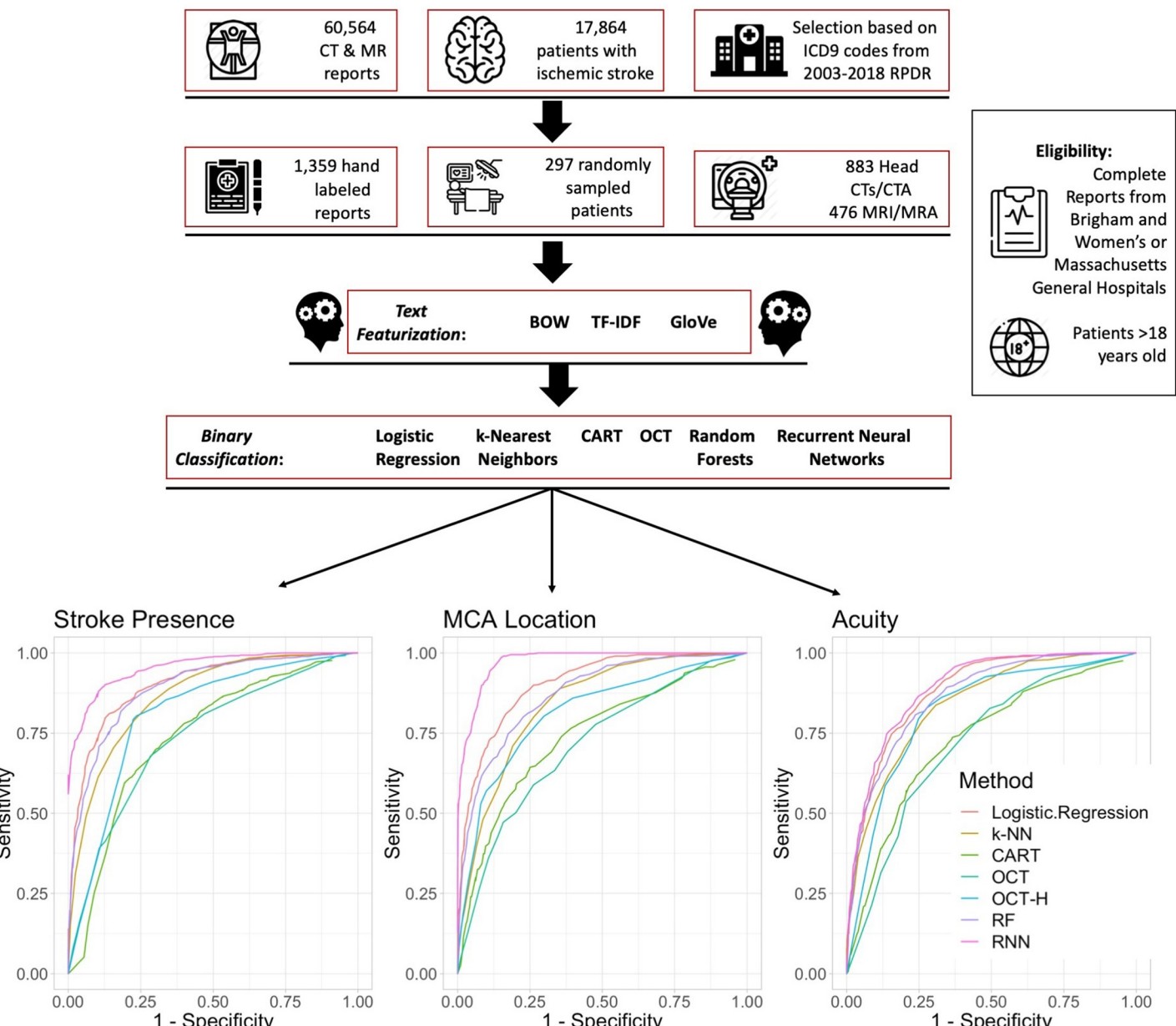

**Fig 1. Flow diagram of Natural Language Processing (NLP) methodology.** Text featurization with GloVe and binary classification lead to Receiver Operator Curves (ROC) for stroke occurrence, MCA location and stroke acuity. Representative ROC curves for each of the text featurization methods are displayed. RPDR = Research Patient Data Registry; CT = Computed Tomography; CTA = Computed Tomography Angiography; MRI = Magnetic Resonance Imaging; MRA = Magnetic Resonance Angiography; BOW = Bag of Words; tf-idf = Term Frequency-Inverse Document Frequency; GloVe = Global Vectors for Word Representation; CART = Classification and Regression Trees; OCT = Optimal Classification Trees; RF = Random Forests; RNN = Recurrent Neural Networks.

and raw images. She found percent agreement for presence of ischemic stroke, MCA location, and acuity were 97%, 95%, and 98%, respectively. The most common cause for discrepancies resolved upon adjudication included small chronic strokes, strokes referred to in the report that were only identified on a prior scan, or subtle early changes that were consistent with symptoms listed in the report and available to the radiologist (S1 Appendix in S1 File).

## Text preprocessing and featurization

To remove basic non-uniformities in unstructured text data, we used the following steps to preprocess radiology reports for further analysis.

1. We removed any incomplete reports, header text (i.e. patient or visit information, procedure details), non-diagnostic standardized language (i.e. names or electronic provider signatures), non-narrative text including "= = = = = =".

2. We converted commonly used word groups, "word tokens", to "n-grams", or a single word group unit without spaces. For example, middle cerebral artery was converted to: "middlecerebralartery".

3. We standardized all whitespace, removed punctuation and converted all text to lowercase.

After preprocessing, narrative text was "featurized" to convert unstructured data into classifiable, numeric information for a machine learning algorithm [8, 9]. We compared simple traditional text featurization methods (BOW, tf-idf) with a recent word embedding technique trained on neurology-specific text. The specific featurization techniques used in our analysis are detailed below and more comprehensively in Table 1 and S1 Appendix in S1 File:

1. Bag of Words (BOW): Bag of words is the simplest model for text featurization, disregarding context, semantic proximity and grammar. Each word, or grouping of words (n-gram) in the main corpus/body of the text is considered a distinct feature. The value of each feature corresponds to the number of times a word was found in a given report.

2. Term Frequency-Inverse Document Frequency (tf-idf): The term frequency-inverse document frequency method (tf-idf) re-weights document features based on the relative importance of the word in the text [2]. Weighting of words is positively correlated to the number of times a word appears in a given document, but is offset by frequency in the training corpus.

3. Global Vectors for Word Representation (GloVe): GloVe is a word-embedding method that quantifies how often pairs of words co-occur in some window in a given text, since these frequencies are likely to have semantic meaning (S1 Fig in S1 File) [4]. For example, the pairs of terms "ice"-"solid" and "steam"-"gas" co-occur much more frequently than pairs "ice"-"gas" and "steam"-"solid." Exact frequencies depend on the specific training set GloVe uses.

## Radiologic stroke featurization training corpus for GloVe

Since standard widely available text corpora do not provide frequent exposure to our concepts of interest (i.e. ischemic stroke), and more specifically the likely co-occurrence of word pairs relevant to stroke, we developed a neurovascular specific corpus to train our GloVe featurization algorithm, including:

1. The complete set of neurology articles on UpToDate™, to capture general neurologically focused medical language [19].

2. Stroke, Pathophysiology Diagnosis and Management, to capture stroke-specific language [20].

3. Yousem's Neuroradiology: The Requisites, to capture neuroradiology specific language [21].

4. A random sample of 10,000 radiology reports from 2010–2017, separate to our testing and training set, to capture language specific to radiology reports of all types.

This training resulted in the first neuroradiology specific set of vector representations, which we made available for other clinical NLP applications and can be found at http://www.mit.edu/~agniorf/files/Glove_Neurology_Embeddings.csv. Our GloVe model parameters included word vector dimension of 100, number of iterations of 50, a window size of 10, and a learning rate of 0.05.

## Report classification

To classify the radiology reports for our three outcomes of interest 1) presence of stroke, 2) stroke location (MCA territory), and 3) stroke acuity, we created predictive models using logistic regression, k-Nearest Neighbors (k-NN), Classification and Regression Trees, (CART) Optimal Classification Trees (OCT) with and without hyperplanes (OCT-H), Random Forest, and Recurrent Neural Networks (RNN) (Table 2; S2 Fig and S1 Appendix in S1 File) [10–16]. Our analysis leverages a wide range of traditional state-of-the-art algorithms including linear regression, tree-based, ensemble, and Neural Network models. The choice of RNN among the various types of NN structures was based on prior research in the NLP field that indicated superior performance when applied to sequential text [16, 22]. RNN coupled with LSTM gates allow for back propagation of information, and thus are able to leverage the order of words in the text [17]. In the derivation cohort, we reported results across a comprehensive combination of all text featurization and predictive techniques outlined above. We performed further external validation using 500 "unseen" reports from an additional medical center, leveraging our two combinations of text featurization techniques and binary classification algorithms. Specifically, we report the performance of interpretable, simple models that use Logistic Regression with BOW and the more complex RNN models coupled with neurology-specific GloVe embeddings.

For validation of our models, we used a grid search and 10-fold cross-validation to select the appropriate values of tuning parameters for all binary classification algorithms. Our parameters for model development included the selection of the regularization term λ, using a maximum of 1000 iterations and a tolerance threshold of 0.01 for logistic regression and the k parameter for the k-NN algorithm from the range of [5, 10, 15, 20]. We selected minimum bucket and maximum depth parameters for tree-based methods across a range of 1–10, and used AUC, entropy, gini, and misclassification accuracy to refine and select the final model. The maximum number of greedy trees for Random Forest was set to 200. Our RNN model used an LSTM network with two hidden layers, including a layer of sentence vectors, and a second layer in which the relations of sentences are encoded in document representation [23]. Further details about the cross-validation process can be found in the S1 Appendix in S1 File.

We trained our models on 75% of the original cohort of 1,359 reports and tested on a withheld test set of 25% for internal validation. For our derivation cohort, we used bootstrapping to randomly split the data five times into training and testing sets. The entire external validation cohort was tested across all five splits of the data. To evaluate model performance on both cohorts, we compared discrimination by reporting Area Under the Curve (AUC) with confidence intervals. We also reported sensitivities, specificities, accuracy, precision, and recall. In the derivation cohort for each prediction task, we report the latter metrics only for the best performing method (GloVe/RNN). For both the internal and external validations, we prioritized sensitivity, and chose a threshold in which sensitivity of >90% produced the highest specificity. For each outcome on our derivation cohort, we evaluated the models' calibration using calibration curves. Moreover, we selected the two best performing classifiers and compared them

**Table 1. Natural language processing and machine learning classifier descriptions.** A. Natural Language Processing Methods. B. Classification Algorithms.

**A**

| | BOW | tf-idf | GloVe |
|---|---|---|---|
| **Methodology** | Each feature represents a word. The value of each feature is the absolute frequency of that word in the document of interest. It can be adapted to include short phrases called n-grams (i.e. mca_stroke; acute_mca_stroke) [8]. | Builds upon BOW by re-weighting the document features based on the relative importance of each word in the text. The relative weight is a function of the term's frequency in the training corpus versus the document of interest [8]. | GloVe constructs word embeddings, representing individual words as d-dimensional vectors. The text is then summarized as the sum of its words. Terms that appear in the same text context are associated with similar representations [4]. |
| | BOW disregards context, semantic proximity and grammar. | tf-idf disregards context, semantic proximity and grammar. | Word vectors better account for grammar and syntax. |
| **Example** | "There is loss of grey-white matter consistent with early ischemic stroke . Impression: There is an ischemic stroke in the MCA distribution." <br><br> "Stroke" = 2, as it is mentioned twice. "Is" also = 2. | tf-idf will reassign the value of the commonly used word " is " given that it occurs so commonly in language. The occurrence of the word " stroke " would be weighted more heavily. | "Ischemic stroke " has a similar vector to "Ischemic infarct " as stroke and infarct are synonymous. These vectors would be assigned similar values. |

**B**

| | Logistic Regression | k-NN | Tree Based Methods | | | Neural Networks |
|---|---|---|---|---|---|---|
| | | | CART | OCT | RF | |
| Description | A statistical model that uses a logistic function to model a binary dependent outcome using one or more inputs [10]. | Classifies each observation by identifying the k most similar observations. The observation is assigned to the class to which the majority of its neighbors belong [11]. | Trains a decision tree with a top-down approach starting at the strongest predictor. Subsequent predictors are used to split the data into smaller classifications. Each leaf (end node) of the tree determines the likelihood of the report belonging to a specific class [12]. | Constructs the best decision tree in a single step, with each split in the tree formed with the knowledge of the other splits. While CART splits one at a time, OCT considers all splits at once [13, 14]. | Builds a large number of CART trees in parallel. A classification prediction is made by averaging the votes of all tree-based models [15]. | A computational model consisting of multiple input layers (each with its own function that informs subsequent layers' predictions) to make a final classification. RNN [16] is a subtype that allows back propagation of information to leverage the order of words in the text [9, 17]. |
| Interpretability | High. | Medium. | High. | High. | Low. | Low. |
| | Potential features are selected by researchers and their independent and adjusted weights and significances can be identified from the model. | Nearest neighbors and their features can be visualized to infer the rationale for classification in low dimensional spaces. | The single decision-tree model allows visualization of the features that lead to splits. | OCTs share the single tree structure in which splits can be followed. | Unlike CART and OCT, RF outcomes are an average of many trees using a random subset of variables. | RNNs are difficult to interpret due to their many layers and bidirectional flow of information. |
| Modifications | Lasso regression assigns a penalty for complex models, resulting in fewer parameters to increase interpretability and reduce over-fitting. | The value of k is selected using cross-validation.* | The value of minimum bucket, maximum depth and complexity parameter is selected via cross-validation. * | Adding hyperplanes to OCT (OCT-H) authorizes multi-variable rather than single-variable splits to improve performance. | The number of trees for each model is selected via cross-validation. | Long Short-Term memory (LSTM) networks is a subclass of RNN that improves the incorporation of time-dependent variables in the model [18]. |

Bag of Words (BOW); Term Frequency-Inverse Document Frequency (tf-idf); Global Vectors for Word representation (GloVe). k-Nearest Neighbors (k-NN); Classification and Regression Trees (CART); Optimal Classification Trees (OCT); Random Forests (RF); Recurrent Networks (RNN).

*Cross-validation: Using different combinations of training set to optimize model parameters.

using the McNemar test [24]. A 2-sided P-value of 0.05 was considered significant. Similar to other NLP studies, we used this test to validate the hypothesis that the two predictive models are equivalent [25]. We report the average performance across all five partitions of the data for

**Table 2. Performance metrics for natural language processing and classification on the derivation cohort.** A) Average AUC metric across all five splits of the data. B) Sensitivity, Specificity, Accuracy and Precision for GloVe Models combined with RNN.

**a)**

| Stroke | | | | | | | |
|---|---|---|---|---|---|---|---|
| Average AUC (95% CI) | **Logistic Regression** | **k-NN** | **CART** | **OCT** | **OCT-H** | **RF** | **RNN** |
| **BOW** | 0.951 (0.943:0.959) | 0.808 (0.767:0.848) | 0.889 (0.868:0.91) | 0.805 (0.774:0.836) | 0.915 (0.899:0.92) | 0.922 (0.902:0.942) | 0.838 (0.811:0.866) |
| **tf-idf** | 0.939 (0.933:0.945) | 0.857 (0.825:0.889) | 0.883 (0.859:0.907) | 0.813 (0.801:0.825) | 0.894 (0.853:0.906) | 0.929 (0.909:0.948) | 0.843 (0.816:0.869) |
| **GloVe** | 0.904 (0.889:0.918) | 0.867 (0.836:0.898) | 0.734 (0.703:0.765) | 0.722 (0.69:0.753) | 0.767 (0.775:0.834) | 0.892 (0.868:0.916) | 0.961 (0.955:0.967) |
| **Location** | | | | | | | |
| Average AUC (95% CI) | **Logistic Regression** | **k-NN** | **CART** | **OCT** | **OCT-H** | **RF** | **RNN** |
| **BOW** | 0.959 (0.944:0.974) | 0.841 (0.816:0.867) | 0.949 (0.93:0.969) | 0.867 (0.838:0.896) | 0.937 (0.919:0.955) | 0.96 (0.943:0.978) | 0.896 (0.873:0.926) |
| **tf-idf** | 0.962 (0.943:0.981) | 0.903 (0.873:0.933) | 0.944 (0.918:0.97) | 0.862 (0.828:0.896) | 0.934 (0.917:0.951) | 0.965 (0.947:0.983) | 0.956 (0.936:0.977) |
| **GloVe** | 0.906 (0.884:0.927) | 0.843 (0.819:0.868) | 0.734 (0.677:0.791) | 0.699 (0.662:0.722) | 0.809 (0.787:0.83) | 0.873 (0.854:0.892) | 0.976 (0.968:0.983) |
| **Acuity** | | | | | | | |
| Average AUC (95% CI) | **Logistic Regression** | **k-NN** | **CART** | **OCT** | **OCT-H** | **RF** | **RNN** |
| **BOW** | 0.898 (0.874:0.922) | 0.815 (0.775:0.854) | 0.797 (0.748:0.846) | 0.735 (0.705:0.764) | 0.797 (0.742:0.852) | 0.901 (0.883:0.919) | 0.754 (0.733:0.779) |
| **tf-idf** | 0.893 (0.865:0.921) | 0.857 (0.826:0.888) | 0.801 (0.762:0.839) | 0.733 (0.703:0.764) | 0.807 (0.764:0.843) | 0.902 (0.876:0.923) | 0.899 (0.875:0.922) |
| **GloVe** | 0.881 (0.842:0.92) | 0.842 (0.805:0.879) | 0.73 (0.684:0.776) | 0.719 (0.66:0.778) | 0.82 (0.766:0.873) | 0.866 (0.824:0.908) | 0.925 (0.894:0.955) |

**b)**

| | Sensitivity | Specificity | Accuracy | Precision | Threshold | | |
|---|---|---|---|---|---|---|---|
| **Stroke** | 0.902 | 0.872 | 0.892 | 0.935 | 0.69 | | |
| **MCA Location** | 0.902 | 0.911 | 0.908 | 0.766 | 0.42 | | |
| **Acuity** | 0.911 | 0.689 | 0.772 | 0.935 | 0.33 | | |

k-Nearest Neighbors (k-NN); Classification and Regression Trees (CART); Optimal Classification Trees (OCT); Random Forests (RF); Recurrent Networks (RNN).

each evaluation criterion. Confidence intervals were calculated for the bootstrapped results. Analysis was performed with R 3.5.2 and Python (scikit-learn, Tensorflow libraries) [26, 27].

## 3. Results

Of 1,359 hand-labeled reports from 297 patients in the derivation cohort, 925 had ischemic strokes, 350 were labeled as "MCA territory" and 522 were labeled as acute. 129 patients were female (43%), and median age at report time was 68 years [IQR 55,79] (S1 Table in S1 File). In the validation cohort, 500 reports were used from 424 patients with a median age of 69 [IQR 59,79] at report time. The sample included 192 female patients (45%). After labeling, 266 reports were classified as strokes, 90 as "MCA territory" and 106 were characterized as acute.

We compared performance of multiple text featurization and classification methods to classify our outcomes of interest (Table 1). For stroke, MCA location, and acuity, we observed best discrimination using our developed GloVe word embedding and RNN classifier algorithm with AUC values of 0.961, 0.976, and 0.925 respectively (Table 2). For simpler tasks, like the

identification of stroke, logistic regression combined with BOW performed comparably to more complex word embedding methods (AUC of 0.951 with Logistic Regression/BOW vs. AUC of 0.961 with GloVe/RNN). However, the difference in discrimination was larger for more nuanced features like acuity (AUC of 0.898 for Logistic Regression/BOW vs. 0.925 for GloVe/RNN). The word embedding approach did not perform as well when paired with logistic regression or single-decision tree methods (Table 2). Receiver Operator Curves (ROCs) are included in Fig 2. We constructed calibration curves for our models, where best performance is represented by a slope of 45˚, and the three best classifiers are included in Fig 3. Random Forest classifiers suffered a decline in calibration, especially in the MCA location task at high predicted probabilities. GloVe/RNN methods appeared to have the best calibration across tasks.

In terms of accuracy, (the fraction of reports from both positive and negative classes that are correctly classified), we found that GloVe/RNN models achieved up to 89 and 91% for stroke presence and MCA location, respectively. Corresponding sensitivities and specificities were both high (0.90 and 0.87 for stroke presence) and (0.90 and 0.91 for MCA location). For the acuity task, while we prioritized sensitivity (0.91), accuracy was less (0.77), reflecting the greater difficulty of this classification (Table 2). Precision/recall and full sensitivities/specificities for GloVe are included (S4 Table and S3 Fig in S1 File).

Finally, we used McNemar's test to compare our best performing GloVe model with the best performing simpler NLP model for each task. Specifically, we compared GloVe/RNN with the second-best performing combination of supervised learning and text featurization technique. For the presence of stroke task, Logistic Regression coupled with BOW had a chi-squared value of 4.79 (p = 0.056). For both the location and acuity outcome, we used the models of tf-idf/RF and showed that both had equivalent performances, 14.78 (p = 0.023) and 26.74 (p = 0.031) respectively. Detailed results for each split of the data are provided (S5 Table in S1 File).

In our external validation cohort, we tested our most sophisticated (GloVe/RNN) and the simplest (BOW/Logistic Regression) methods. We found that BOW/Logistic Regression (AUCs 0.89, 0.86 and 0.80 respectively for stroke, location, acuity) did not generalize as well as GloVe/RNN (AUC 0.92, 0.89, 0.93) in the external population (Table 3). We continued to prioritize sensitivities in the external validation population for GloVe/RNN (0.90–0.92), and specificities were decreased for stroke and MCA location (0.75, 0.70) compared to the internal validation population (0.87, 0.91). Specificities remained the same (0.69) for the acuity task.

## Discussion

Accurate automated information extraction will be increasingly important as more medical researchers, hospital systems, and academic institutions leverage "big data" from electronic medical records. Unlike structured, discrete data like laboratory values or diagnoses codes, unstructured text is challenging to analyze. However, clinicians frequently record essential observations, interpretations, and assessments that are otherwise absent from the remainder of the medical record. In order to fully leverage our ability to access such data through the medical record, we must have validated methods to extract meaningful information. Specific to radiology reports, there are several important applications of accurate automated extraction of information through NLP. Automatic, real-time identification of specific subpopulations (such as patients with acute MCA stroke) can improve clinical workflow and management by triaging eligible patients to timely treatments or higher levels of care [3]. NLP approaches can facilitate research by identifying both populations (i.e. patients with stroke, tumor or aneurysms) and outcomes (i.e. presence of hemorrhagic conversion or edema) more feasibly than

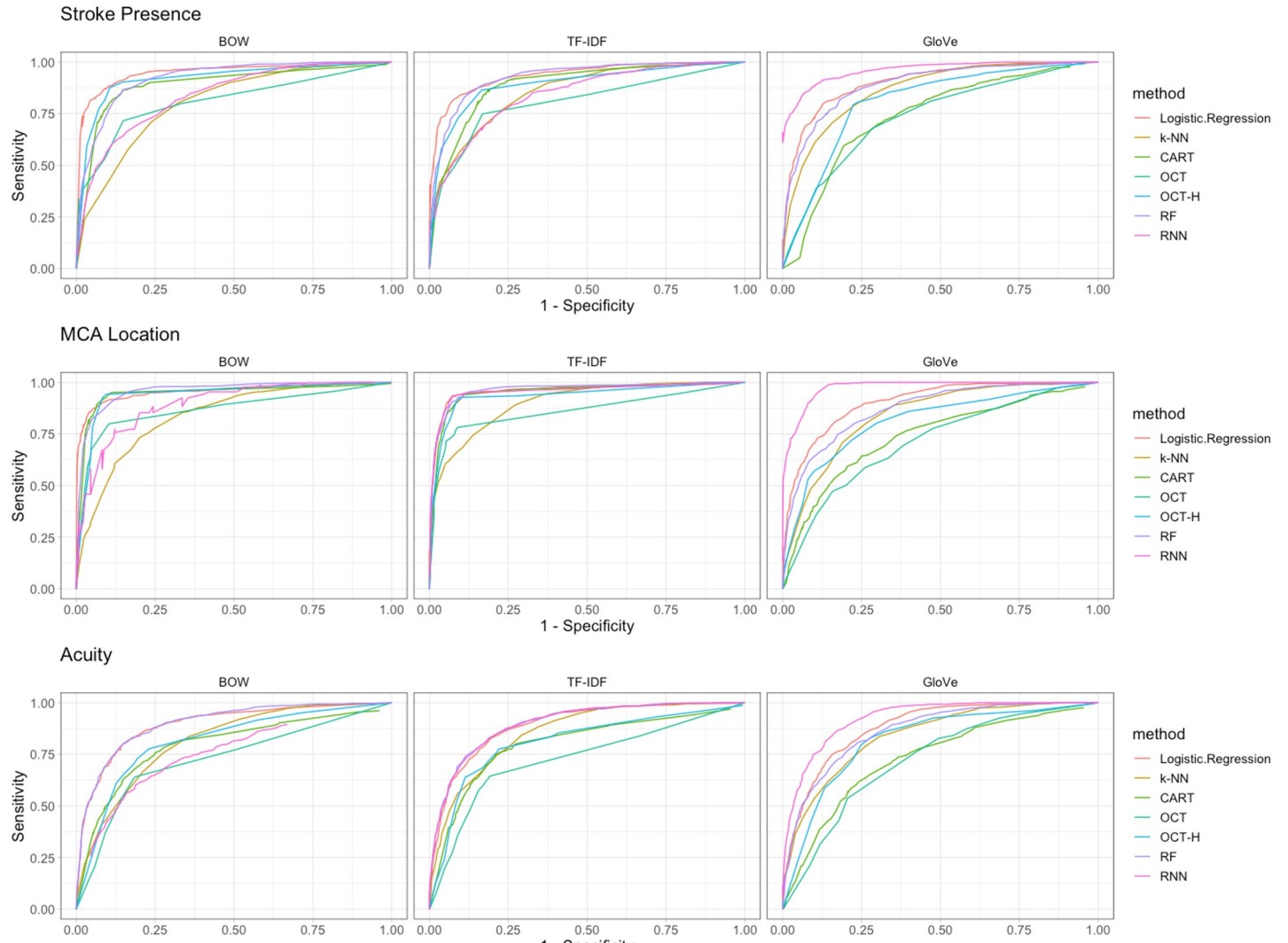

**Fig 2. Receiver operating curves for NLP classification.** A, stroke presence; B, MCA location; C, acuity. These curves represent different combinations of text featurization (BOW, tf-idf, GloVe) and binary classification algorithms (Logistic Regression, k-NN, CART, OCT, OCT-H, RF, RNN). GloVe and RNN achieved the highest AUC for all three tasks (>90%). Similar results were achieved for simple tasks by BOW or tf-idf paired with Logistic Regression. The results presented average the mean sensitivity and specificity over five random splits of the data. In a ROC curve the true positive rate (Sensitivity) is plotted as a function of the false positive rate (1-Specificity) for different cut-off points of a parameter. Each point on the ROC curve represents a sensitivity/specificity pair corresponding to a particular decision threshold. The area under the ROC curve (AUC) is a measure of how well a parameter can distinguish between the two subpopulation groups.

manual review, and potentially more accurately than billing codes. Indeed, of our 1,359 radiographic reports derived from patients with billing codes of stroke, only 925 (68%) had a radiographically reported ischemic stroke, which raises the question as to whether NLP can assist in improving diagnostic classification. In this study, we developed a comprehensive framework to create a vector-based NLP method specifically targeted to identify stroke features from unstructured text. We then tested the ability of multiple machine learning methods to classify specific stroke features and compared performance. We designed our study to identify these three tasks separately as opposed to a single task ("acute middle cerebral artery stroke") because our objective was to create an NLP identification system that can be expanded to multiple stroke types in the future.

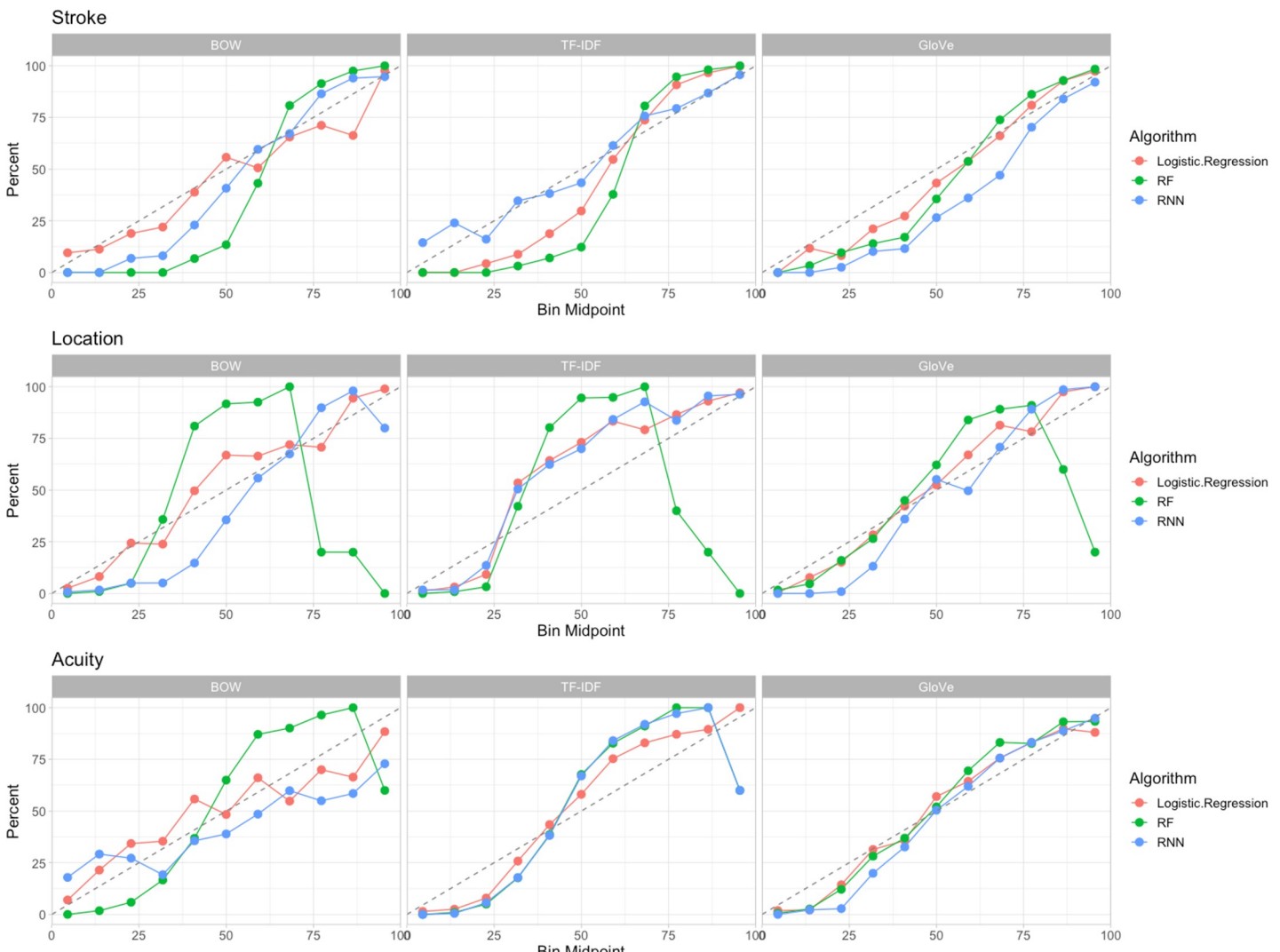

**Fig 3. Calibration curves for NLP classification.** A, stroke presence; B, MCA location; C, acuity. These curves represent different combinations of text featurization (BOW, tf-idf, GloVe) and binary classification algorithms (Logistic Regression, RF, RNN). We created plots showing the relation between the true class of the samples and the predicted probabilities. We binned the samples according to their class probabilities generated by the model. We defined the following intervals: [0,10%], (10,20%], (20,30%], . . . (90,100%]. We subsequently identified the event rate of each bin. For example, if 4 out of 5 samples falling into the last bin are actual events, then the event rate for that bin would be 80%. The calibration plot displays the bin mid-points on the x-axis and the event rate on the y-axis. Ideally, the event rate should be reflected as a 45˚ line. The results presented are aggregated over five different splits of the data. We show results of the three best performing methods in each task.

We found that NLP methods perform well at extracting featurized information from radiology reports (AUCs >0.9 for all three tasks). True to our hypothesis, word-embedding methods like GloVe improved overall accuracy of feature identification, especially when paired with deep learning methods like RNN, which are less interpretable (harder to distinguish features contributing to performance) than simpler classification algorithms like logistic regression or single-decision trees. However, RNN's have been particularly successful in NLP applications, where the sequence of words in the text can crucially alter the overall meaning of the corpus [9]. Because the field of NLP is rapidly expanding, variations of featurization methods are used and trialed for different purposes. We chose to use BOW, tf-idf and GloVe because they were representative of the simplest, the most frequently used, or an innovative word-embedding approach that better captures semantic meaning, respectively.

**Table 3. Performance metrics for natural language processing and classification on the validation cohort across all outcomes for BOW with logistic regression and RNN with GloVe.** a) Average AUC metric across all five splits of the data. b) Sensitivity, Specificity, Accuracy and Precision for GloVe Models combined with RNN on the BMC Validation Cohort.

a)

|  | Stroke | Location | Acuity |
|---|---|---|---|
| BOW+Log.Reg | 0.892 (0.875:0.91) | 0.857 (0.845:0.869) | 0.797 (0.768:0.828) |
| GloVe+RNN | 0.920.908:0.932) | 0.893.88:0.905) | 0.925.906:0.946) |

b)

|  | Sensitivity | Specificity | Accuracy | Precision |
|---|---|---|---|---|
| Stroke | 0.915 | 0.752 | 0.828 | 0.764 |
| MCA Location | 0.898 | 0.7 | 0.862 | 0.932 |
| Acuity | 0.914 | 0.689 | 0.866 | 0.916 |

We acknowledge that there are various widely accepted word embedding techniques, such as Word2Vec, the Distributed memory (DM)-document vector (DV) model, the continuous bag of words (cBoW) model, the continuous skip-gram (CSG) model, and FastText [1, 5, 28]. Recently, investigators also proposed a hybrid method, called Intelligent Word Embedding (IWE), that combines semantic-dictionary mapping and a neural embedding technique for creating context-aware dense vector representation of free-text clinical narratives [29]. However, our aim was to demonstrate whether a neurology-specific embedding model could improve upon simpler techniques that do not consider context and semantic meaning in their word representations. Given the significant computational resources required for the creation of the embeddings and prior research demonstrating equivalence between the algorithms' objectives, we limited our analysis to one word embedding technique [30]. We chose to use GloVe because this approach outperformed other word-embedding methods, and has been shown to do so with smaller training sets, which is important when considering how our contributions may be applied to other investigators for research and/or clinical use [4].

This investigation is part of a wider literature that employs deep learning in clinical NLP [31]. In this study, we employ a specific RNN structure that had been previously and successfully used in combination with GloVe embeddings [16]. An increasing number of deep learning structures are being employed in similar applications such as autoencoders [32, 33], deep belief networks [34], memory residual NN [35], and attention mechanisms like BERT [36]. Future research directions could focus on leveraging these other NLP structures with neurology-specific embeddings and comparing their performance.

Our work is consistent with other studies reporting simple methods like BOW are suitable for extracting unstructured text information. One group found that BOW paired with lasso logistic regression had high performance (AUCs of >0.95) for critical head CT findings [1]. Kim and colleagues' found that a single decision tree outperformed more complicated support vector machines in identifying acute ischemic stroke on MRIs [37]. Garg and colleagues used various machine learning methods to classify stroke subtype from radiology reports and other unstructured data [38]. They achieved a kappa of 0.25 using radiology reports alone, which improved to 0.57 when they used combined data. In our study, our GloVe embedded vector approach was specifically tailored for the detection of vascular neurologic disorders, and outperformed other methods in correctly classifying stroke acuity, particularly when paired with a neural network structure. Additional analysis also demonstrated that general purpose embeddings such as the ones trained only on Wikipedia provide significantly lower performance (S6 Table in S1 File). Namely, an RNN classifier achieved 0.74 (0.70:0.75) AUC for presence, 0.75 (0.72:0.79) AUC for location, and 0.693 (0.61:0.73) AUC for acuity of stroke—a decrease of at

least 0.2 in discriminatory performance compared to our proposed embeddings. These results emphasize the need for radiographic-specific word representations that capture the semantic relations of medical vocabulary. Because RNNs account for word order, we expect these methods will be increasingly used for accurate natural language processing of medical text data.

### Limitations

There are several important limitations to our work. Similar to other studies, our radiology corpus consisted of reports from only two hospitals, which may reduce our generalizability in other systems. Also, the use of both computed tomography and magnetic resonance imaging reports increases heterogeneity for model development; however, given the finite number of ways in which reports describes stroke characteristics regardless of imaging modality, we sought to test a method that could be widely applied to radiographic text.

### Strengths and future directions

Strengths of our study include the development of a tailored word-embedding approach to vascular neurologic disorders, the development of multiple models testing the optimal combination of NLP and classification algorithms, generalizability to both CT scans and MRIs, its external validation in a racial-ethnic and socio-economically diverse cohort, and the ability to expand this framework to additional stroke characteristics (increased locations, hemorrhagic conversion). While our word-embedding approach was specifically tailored to neurovascular disorders, similar approaches could be used to generate word vectors for other disease states, including oncology and cardiology. Moreover, while our data extraction of unstructured text focused on radiology reports, further work in this area could assist in the retrieval of essential information in progress notes, and interrogation of discrepancies in the medical record that result from "copy/paste." As we gather more electronic data on patients, easy information retrieval will become increasingly important as a strategy to scale research and improve quality.

## 4. Conclusion

Automated machine learning methods can extract diagnosis, location and acuity of stroke with high accuracy. Word-embedding approaches and RNNs achieved the best performance in correct classification of stroke and stroke characteristics. Our results provide a framework for expeditiously identifying salient stroke features from radiology text that can triage high-risk imaging findings and identify patient populations of interest for research. Future directions include improving performance through the study of hybrid rule-based and machine learning methods. Work in this area is particularly important as accurate, accessible methods to automate data extraction will become increasingly relevant for academic, tertiary, and non-tertiary centers who aim to improve clinical, administrative, and quality care.

## Supporting information

**S1 File.**
(DOCX)

## Acknowledgments

We would like to thank the anonymous reviewers from PLOS ONE for their insightful comments that significantly improved the paper.

## Author Contributions

**Conceptualization:** Charlene Jennifer Ong, Agni Orfanoudaki, Rebecca Zhang, Stelios Smirnakis, Dimitris Bertsimas.

**Data curation:** Charlene Jennifer Ong, Agni Orfanoudaki, Rebecca Zhang, Meghan Hutch, Brenton Prescott.

**Formal analysis:** Agni Orfanoudaki, Rebecca Zhang, Francois Pierre M. Caprasse.

**Funding acquisition:** Charlene Jennifer Ong, Meghan Hutch, Stelios Smirnakis, Dimitris Bertsimas.

**Investigation:** Charlene Jennifer Ong, Agni Orfanoudaki, Rebecca Zhang, Liang Ma, Darian Fard, Oluwafemi Balogun, Matthew I. Miller, Margaret Minnig, Hanife Saglam.

**Methodology:** Charlene Jennifer Ong, Agni Orfanoudaki, Rebecca Zhang, Stelios Smirnakis.

**Project administration:** Charlene Jennifer Ong, Oluwafemi Balogun, Hanife Saglam, Dimitris Bertsimas.

**Resources:** Charlene Jennifer Ong, Dimitris Bertsimas.

**Software:** Agni Orfanoudaki, Rebecca Zhang, Francois Pierre M. Caprasse.

**Supervision:** Charlene Jennifer Ong, Meghan Hutch, Dimitris Bertsimas.

**Validation:** Charlene Jennifer Ong, Agni Orfanoudaki, Meghan Hutch, Brenton Prescott.

**Visualization:** Charlene Jennifer Ong, Agni Orfanoudaki, Meghan Hutch, Darian Fard.

**Writing – original draft:** Charlene Jennifer Ong, Agni Orfanoudaki, Darian Fard, David M. Greer.

**Writing – review & editing:** Charlene Jennifer Ong, Agni Orfanoudaki, Brenton Prescott, David M. Greer, Stelios Smirnakis, Dimitris Bertsimas.

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
