## [Decision Letter · Decision Letter 0]

6 Jan 2020

PONE-D-19-31481

Novel Natural Language Processing techniques using Machine Learning methods to identify ischemic stroke, acuity and location from radiology reports

PLOS ONE

Dear Dr Ong,

Thank you for submitting your manuscript to PLOS ONE. After careful consideration, we feel that it has merit but does not fully meet PLOS ONE’s publication criteria as it currently stands. Therefore, we invite you to submit a revised version of the manuscript that addresses the points raised during the review process.

We would appreciate receiving your revised manuscript by Feb 20 2020 11:59PM. To enhance the reproducibility of your results, we recommend that if applicable you deposit your laboratory protocols in protocols.io, where a protocol can be assigned its own identifier (DOI) such that it can be cited independently in the future. For instructions see: http://journals.plos.org/plosone/s/submission-guidelines#loc-laboratory-protocols

We look forward to receiving your revised manuscript.

Kind regards,

Yifan Peng, Ph.D.

Academic Editor

PLOS ONE

Journal Requirements:

Additional Editor Comments (if provided):

Please also double check the URL and Github link.

Reviewers' comments:

Reviewer's Responses to Questions

**Comments to the Author**

1. Is the manuscript technically sound, and do the data support the conclusions?

Reviewer #1: Yes

Reviewer #2: Partly

2. Has the statistical analysis been performed appropriately and rigorously? 

Reviewer #1: Yes

Reviewer #2: No

3. Have the authors made all data underlying the findings in their manuscript fully available?

Reviewer #1: Yes

Reviewer #2: No

4. Is the manuscript presented in an intelligible fashion and written in standard English?

Reviewer #1: Yes

Reviewer #2: Yes

5. Review Comments to the Author

Reviewer #1: Thanks for the opportunity reviewing this interesting manuscript. The authors evaluated both traditional machine learning algorithms and deep learning algorithms with several common features for radiology reports-related NLP tasks. The manuscript is well written and easy to follow. While I appreciate authors’ efforts in building annotated datasets and an extensive set of experiments, I feel like additional justifications and experiments should be provided to make this study more robust. Here are some comments the authors could consider to further improve this manuscript:

Major comments:

1. The first word in the title is “novel”. However, throughout the manuscript, the authors did not provide justifications of the novelty of their methods. To my knowledge, the algorithms and features the authors evaluated in the present study are pretty routine. The authors need to justify the novelty of their study or remove the word “novel” in their title.

2. The authors need carefully check the second paragraph in the introduction. The definitions of some NLP-related concepts are not rigorous. For example, non-machine learning-based approach can also build NLP systems; text featurization can also converts text into features using non-machine learning-based methods; GloVe is only one type of word embedding algorithms, perhaps not the most popular one. In addition, the first few sentences in the second paragraph seems to be related the general introduction of NLP, however, the sentences since “After converting language into relevant” are related to radiology-specific NLP. The authors should consider to adding some transitions in the between.

3. The justification of choosing the algorithms is not very clear. For example, why choose GloVe over other embedding algorithms, including word2vec, fastText? Why to choose RNN over CNN?

4. The implementations of some algorithms are not very clear. What are the hyper-parameters of RNN? What are the major parameters for the training of GloVe embedding, including dimension, number of iterations, window size, etc?

5. I feel like additional experiments are needed to be done to make the study more robust. For example, the author trained GloVe vector on the radiology reports. To demonstrate its superiority, the authors should also at least test random embedding or pre-trained GloVe em-bedding from general domain. The authors don’t have to report the results in the manuscript, but relevant discussion is necessary.

6. In the last two years, we’ve witnessed the booming trends of using contextualized embed-ding (e.g. ELMo), BERT and other deep learning architectures in clinical NLP. Some relevant discussion should be provided.

a. Wu S, et al. Deep learning in clinical natural language processing: a methodical re-view. Journal of the American Medical Informatics Association. 2019 Dec 3.

Minor comments:

1. “TF-IDF” is “term frequency–inverse document frequency”, not “Text Featurization-Inverse Document Frequency”

2. Please remove the last dot (.) in page 11, line 162. The current URL can’t be accessed.

3. An overall description of the label corpus, including the distribution of different labels, length of text, etc., should be provided

Reviewer #2: Authors present a comparison study where some text featurizations and standard classification methods have been compared against each other on the same test dataset.

The manuscript is easy to read but the technical contribution is limited.

Author may consider the following points for improvement -

1. Author mentioned that RNN provides the best performance. But, 1,359 reports are too few to train a standard RNN model. May be they are getting this performance as the model is overfitted to their dataset. Author should mention the number of trainable parameters in the network and also try to test with an external test cohort (not from the same institution).

2. Given that all the models' performance is over 0.8, may be the task is also simple for the ML models. Did author try to implement a simple rule-based system with Negation and uncertainty detection? Author should also reports the inter-annotator agreement value.

3. Quality of the figure text is very low and difficult to read.

4. There are also few hybrid NLP models [1] available that combines rule-based with machine learning. Author may also extend their experiment with this kind of techniques.

[1] https://www.sciencedirect.com/science/article/pii/S1532046417302575

---

## [Author Response · Author response to Decision Letter 0]

27 Mar 2020

We have responded to all reviewer and editorial comments in the "Response to Reviewers.docx" document. See attached in the submission. A copy of our responses can also be found below.

We thank the reviewers for their critical assessment of our work. In response, we have made the following modifications which we believe has improved the quality substantially. In particular, the key changes that we have made to this paper are:

• We have labeled and performed external validation of our methods on 500 radiology reports from an additional hospital (Boston Medical Center). This medical center has a socioeconomic and racially-ethnic diverse cohort.

• To compare how our Vascular neurologic trained cohort compared with generally trained word vectors, we trained new Recurrent Neural Network models using pre-trained GloVe embeddings from Wikipedia text only.

• We inquired about permissions for data access. After discussion with our local IRB, they came to the determination that free-text raw data used to develop our algorithms contained protected health/patient-sensitive information including names, birthdates and dates of diagnostic tests, which could not be adequately redacted, given the size of the dataset, and that they may occur in multiple areas of the document. Therefore, we were advised that there were ethical and legal restrictions on sharing our complete data-set. We have provided a letter from our IRB committee with their name and contact information. We have included our code, examples of redacted free text, and summary statistics including demographics, age, and race. Our data access sharing plan includes permissible access using a data sharing agreement and would require contacting the IRB (contact information provided here and in the included letter: Megan Morash: Partners Human Research Committee 399 Revolution Drive - Suite 710, Somerville, MA 02145 Tel: (857) 282-1900, Fax: (857) 282-5693)

• We created additional explanatory tables that provide further information regarding summary statistics for both the derivation and validation population of reports.

• We included detailed information regarding the parameter tuning and selection process for all Machine Learning techniques.

• We updated our figures to higher resolution and addressed editorial comments regarding the quality of the manuscript.

In the following, we address the reviewers’ comments point-by-point:

Additional Editor Comments:

1. Please ensure your manuscript meets PLOS One’s style requirement including those for file naming. 

We have revised our file names to be consistent with PLOS One’s style requirement. 

2. Please also double check the URL and Github link.

We have amended the link to the Neurology-specific GloVe embeddings in the Methods Section. We have also included the same reference in the Data Availability Statement as well as a link to our GitHub repository. In the latter, future readers have access to the code we used in our analysis.

Reviewer Comments to the Author

Reviewer 1 

Thanks for the opportunity reviewing this interesting manuscript. The authors evaluated both traditional machine learning algorithms and deep learning algorithms with several common features for radiology reports-related NLP tasks. The manuscript is well written and easy to follow. While I appreciate authors’ efforts in building annotated datasets and an extensive set of experiments, I feel like additional justifications and experiments should be provided to make this study more robust. 

Thank you for your compliments and consideration of this paper. The suggested supplementary analysis significantly improved both the content and presentation quality of our work. Below you will find our responses to the suggestions.

Major comments:

1. The first word in the title is “novel”. However, throughout the manuscript, the authors did not provide justifications of the novelty of their methods. To my knowledge, the algorithms and features the authors evaluated in the present study are pretty routine. The authors need to justify the novelty of their study or remove the word “novel” in their title.

We amended the title of our paper to be: “Machine Learning and Natural Language Processing methods to identify ischemic stroke, acuity, and location from radiology reports.” In our work, we leverage state-of-the-art as well as traditional supervised and unsupervised machine learning techniques in the neurology setting. Our contributions involve a comprehensive framework for feature extraction from radiographic text that can be generalized to other outcomes of interest, including novel neurology-specific word embeddings.

2. The authors need carefully check the second paragraph in the introduction. The definitions of some NLP-related concepts are not rigorous. For example, non-machine learning-based approach can also build NLP systems; text featurization can also converts text into features using non-machine learning-based methods; GloVe is only one type of word embedding algorithms, perhaps not the most popular one. In addition, the first few sentences in the second paragraph seems to be related the general introduction of NLP, however, the sentences since “After converting language into relevant” are related to radiology-specific NLP. The authors should consider to adding some transitions in the between.

We altered the wording in the Introduction (Lines 78-89) to address these concerns. Specifically, we provide a broader definition for Natural Language Processing and explain that we will focus in particular on Machine Learning models for text featurization. Moreover, we clarify that GloVe is an example of a word embedding technique along with Word2Vec. In addition, we changed the sequence of text in the second paragraph to enable a smoother transition from a general NLP introduction to the radiographic-text application.

3. The justification of choosing the algorithms is not very clear. For example, why choose GloVe over other embedding algorithms, including word2vec, fastText? Why to choose RNN over CNN?

We justify the selection of GloVe over other embedding algorithms in the Discussion section (Lines 387-400). A respective explanation for the choice of RNN over CNN is provided in the “Report Classification” section of Methods (Lines 229-234).

4. The implementations of some algorithms are not very clear. What are the hyper-parameters of RNN? What are the major parameters for the training of GloVe embedding, including dimension, number of iterations, window size, etc?

We provide details regarding the tuning of the hyperparameters across all Machine Learning methods in both the Methods section of the Manuscript as well as in the Supplementary Data eAppendix Sections 3 and 4, including (list dimension, number of iterations and window size) along with their description.

5. Additional experiments are needed to be done to make the study more robust. For example, the author trained GloVe vector on the radiology reports. To demonstrate its superiority, the authors should also at least test random embedding or pre-trained GloVe embedding from general domain. The authors don’t have to report the results in the manuscript, but relevant discussion is necessary.

Following the reviewer’s suggestion, we performed additional analysis where we used pre-trained GloVe embeddings with Recurrent Neural Networks (LSTM) to predict presence, location, and acuity of ischemic stroke. We limited our experiments only to the RNN classifier since this is the only supervised learning method that had superior performance when leveraging word embeddings. We provide a detailed table of our results in the Supplementary Data (eTable 6) and comment on the results in the Discussion. We leveraged the 300-dimensional version, trained on Wikipedia text with 6B tokens and uncased vocabulary of 400K words available here (https://nlp.stanford.edu/projects/glove/). We followed a similar training procedure to the one applied for all other experiments. We replaced out-of-vocabulary words with a randomly initialized UNK token to represent unknown words. This technique was used to represent words, such as encephalomalacia, that were not included in the available Wikipedia-based embeddings. 

6. In the last two years, we’ve witnessed the booming trends of using contextualized embed-ding (e.g. ELMo), BERT and other deep learning architectures in clinical NLP. Some relevant discussion should be provided.

We have included a new section in the Discussion (Lines 401-407) where we comment on these recent techniques, comparing their approach with our suggested framework, including Wu et al’s recent work.

Minor comments:

1. “TF-IDF” is “term frequency–inverse document frequency”, not “Text Featurization-Inverse Document Frequency”

2. Please remove the last dot (.) in page 11, line 162. The current URL can’t be accessed.

We thank the reviewer for noticing these errors and have corrected them from the original submission.

3. An overall description of the label corpus, including the distribution of different labels, length of text, etc., should be provided.

We have included this information in eTable 1 of Supplementary Data.

Reviewer 2

1. Author mentioned that RNN provides the best performance. The authors should mention the number of trainable parameters in the network and also try to test with an external test cohort (not from the same institution).

We performed additional external validation of our models using 500 radiology reports from Boston Medical Center. As the largest safety-net hospital in New England, Boston Medical Center has a markedly different population than MGH and Brigham and Women’s. It serves a diverse racial-ethnic and socioeconomic population. Approximately 60% of its patients are black or Hispanic, as compared to 4% of MGH and Brigham and Women’s Hospitals. Therefore, we felt that external validation on this cohort would be a good test of the generalizability of our model. We applied the same inclusion criteria as the derivation cohort. We found that our predictive accuracy is comparable to the one of the testing set on the Massachusetts General and Brigham and Women’s Hospitals. We provide a detailed description of the population in the Methods section and an analysis of our performance in the Results section. We mention the number and ranges of the trainable parameters including (batch size, minimum bucket, maximum depth, etc.) across all ML algorithms in the Methods Section as well as in the eAppendix of Supplementary Data. 

2. Given that all the models' performance is over 0.8, may be the task is also simple for the ML models. Did author try to implement a simple rule-based system with Negation and uncertainty detection? Author should also report the inter-annotator agreement value.

We report the inter-annotator agreement value both in the Methods section (Lines 157-168) as well as in the Supplementary Data in eTables 2-3. Our project’s initial objective was focused on timely prediction of cerebral edema for acute MCA stroke patients. Our initial approach involved a simple rule-based system where we looked up key words, such as stroke, acute, infarct, etc. However, we quickly realized this technique resulted in many false positives and false negatives. We elaborated our efforts by including negation and n-gram detection. Nevertheless, our report identification discrimination performance did not reach results upon 75%. Thus, we decided to employ ML and NLP methods to correctly identify the appropriate population for our study.

3. Quality of the figure text is very low and difficult to read.

We have updated our figures to improve the resolution and quality of the image and render the text easier to read.

4. There are also few hybrid NLP models available that combine rule-based with machine learning. Author may also extend their experiment with this kind of techniques.

This is an interesting approach for text featurization in the clinical setting that we acknowledge in our discussion. We believe that it could be an interesting alternative that combines semantic-dictionary mapping with NN based-approaches such as Word2Vec. Our aim in this work was to investigate an efficient and accurate way to identify stroke patients from radiographic text. We show that a neurology-specific embedding model could improve simpler techniques that do not consider context and semantic meaning in their word representations and lead to performance that reaches 93-97%. Given the significant computational resources required for the creation of the embeddings and prior research demonstrating equivalence between the algorithms’ objectives, we limit our analysis to one word embedding technique. However, this is an excellent area for further study and we will consider this approach for a future manuscript.

---

## [Decision Letter · Decision Letter 1]

3 Jun 2020

PONE-D-19-31481R1

Machine Learning and Natural Language Processing Methods to Identify Ischemic Stroke, Acuity and Location from Radiology Reports

PLOS ONE

Dear Dr. Ong,

Thank you for submitting your manuscript to PLOS ONE. After careful consideration, we feel that it has merit but does not fully meet PLOS ONE’s publication criteria as it currently stands. Therefore, we invite you to submit a revised version of the manuscript that addresses the points raised during the review process.

We look forward to receiving your revised manuscript.

Kind regards,

Yifan Peng, Ph.D.

Academic Editor

PLOS ONE

Reviewers' comments:

Reviewer's Responses to Questions

**Comments to the Author**

1. If the authors have adequately addressed your comments raised in a previous round of review and you feel that this manuscript is now acceptable for publication, you may indicate that here to bypass the “Comments to the Author” section, enter your conflict of interest statement in the “Confidential to Editor” section, and submit your "Accept" recommendation.

Reviewer #1: All comments have been addressed

2. Is the manuscript technically sound, and do the data support the conclusions?

Reviewer #1: Yes

3. Has the statistical analysis been performed appropriately and rigorously? 

Reviewer #1: N/A

4. Have the authors made all data underlying the findings in their manuscript fully available?

Reviewer #1: Yes

5. Is the manuscript presented in an intelligible fashion and written in standard English?

Reviewer #1: Yes

6. Review Comments to the Author

Reviewer #1: Thanks for the revision. Most of my comments have been addressed.

Some minor comments:

1. Line 82: “NLP involves” to “ML-based NLP involves”

2. Line 83: remove “using specifically tailored machine learning methods”. Many featurization methods are not ML-based. For example, bag-of-words, TFIDF, etc.

3. Line 85: “tf-idf” to “TF-IDF”

4. Line 86-87: “learn vector representations of word relationships” to “learn a distributed representation for words”

---

## [Author Response · Author response to Decision Letter 1]

4 Jun 2020

We thank the reviewers for their assessment of our work. We have addressed the minor changes suggested in the last revision of our investigation.

In the following, we address the reviewers’ comments point-by-point:

Reviewer Comments to the Author

Reviewer 1 

1. Line 82: “NLP involves” to “ML-based NLP involves”

We have adjusted the wording accordingly.

2. Line 83: remove “using specifically tailored machine learning methods”. Many featurization methods are not ML-based. For example, bag-of-words, TFIDF, etc.

We have removed the highlighted expression. 

3. Line 85: “tf-idf” to “TF-IDF”

The abbreviation has been adjusted.

4. Line 86-87: “learn vector representations of word relationships” to “learn a distributed representation for words”

We have adjusted the expression based on the reviewer’s recommendation.

---

## [Editor Report · Decision Letter 2]

5 Jun 2020

Machine Learning and Natural Language Processing Methods to Identify Ischemic Stroke, Acuity and Location from Radiology Reports

PONE-D-19-31481R2

Dear Dr. Ong,

We’re pleased to inform you that your manuscript has been judged scientifically suitable for publication and will be formally accepted for publication once it meets all outstanding technical requirements.

Kind regards,

Yifan Peng, Ph.D.

Academic Editor

PLOS ONE
---

## [Editor Report · Acceptance letter]

10 Jun 2020

PONE-D-19-31481R2 

Machine Learning and Natural Language Processing Methods to Identify Ischemic Stroke, Acuity and Location from Radiology Reports 

Dear Dr. Ong:

I'm pleased to inform you that your manuscript has been deemed suitable for publication in PLOS ONE. Congratulations! Your manuscript is now with our production department. 

Kind regards, 

on behalf of

Dr. Yifan Peng 

Academic Editor

PLOS ONE